# Comparisons between Plant and Animal Stem Cells Regarding Regeneration Potential and Application

**DOI:** 10.3390/ijms24054392

**Published:** 2023-02-23

**Authors:** Lulu Liu, Lu Qiu, Yaqian Zhu, Lei Luo, Xinpei Han, Mingwu Man, Fuguang Li, Maozhi Ren, Yadi Xing

**Affiliations:** 1Zhengzhou Research Base, State Key Laboratory of Cotton Biology, School of Agricultural Sciences, Zhengzhou University, Zhengzhou 450001, China; 2State Key Laboratory of Cotton Biology, Institute of Cotton Research, Chinese Academy of Agricultural Sciences, Anyang 455000, China; 3School of Pharmaceutical Sciences (Shenzhen), Shenzhen Campus, Sun Yat-sen University, Shenzhen 518107, China; 4Hainan Yazhou Bay Seed Laboratory, Sanya 572000, China; 5Institute of Urban Agriculture, Chinese Academy of Agricultural Sciences, Chengdu 610000, China

**Keywords:** plant regeneration, animal regeneration, stem cell, molecular mechanism, regeneration applications

## Abstract

Regeneration refers to the process by which organisms repair and replace lost tissues and organs. Regeneration is widespread in plants and animals; however, the regeneration capabilities of different species vary greatly. Stem cells form the basis for animal and plant regeneration. The essential developmental processes of animals and plants involve totipotent stem cells (fertilized eggs), which develop into pluripotent stem cells and unipotent stem cells. Stem cells and their metabolites are widely used in agriculture, animal husbandry, environmental protection, and regenerative medicine. In this review, we discuss the similarities and differences in animal and plant tissue regeneration, as well as the signaling pathways and key genes involved in the regulation of regeneration, to provide ideas for practical applications in agriculture and human organ regeneration and to expand the application of regeneration technology in the future.

## 1. Introduction

Animals and plants are subjected to a variety of stimuli during their life span that can cause tissue damage. Both animals and plants promote tissue regeneration through adult stem cells or by the induction of stem cell differentiation to maintain their lives [1]. Tissue regeneration refers to the continuous renewal of biological tissues, the re-differentiation of existing adult tissues to produce new organs, or the repair process after tissue damage. It is one of the phenomena of biological life [2,3]. As plants are sessile, they face various challenges in the external environment. Both lower and higher plants have dramatic regenerative capacities. The super-regenerative capacity of plants is important for maintaining their survival [4]. The regenerative capacity of animals is species-specific. For example, planarians can regenerate whole bodies from tissue fragments of almost any part of the body [5,6,7]. Amphibians such as salamanders can also completely regenerate lost organs and limbs, such as the legs, gills, tail, retina, spinal cord, and heart [8,9]. Although the zebrafish is a vertebrate, it has dramatic regenerative capacity and is, therefore, often used as a model of organ regeneration. Zebrafish can regenerate their hearts, livers, spinal cords, and caudal fins [10,11,12]. Humans, however, can only regenerate intestinal cells, skin, and bones, either continuously or periodically [13,14].

Regeneration of animals and plants is dependent upon stem cells. Stem cells undergo differentiation and division to form the tissues or organs required by animals and plants. Plant stem cells mainly exist in the meristem, upon which the formation of plant organs is reliant [15,16,17]. The existence of meristems ensures plasticity in the growth and development of plants [18]. Plant regeneration is mainly regulated by auxin and cytokinin signaling [19]. In animals, Wnt/β-catenin, Hedgehog (Hh), Hippo, Notch, Bone Morphogenetic Protein (BMP), Transforming growth factor-beta (TGF-β), and other signaling pathways regulate animal tissue regeneration [20,21]. Interestingly, the target of rapamycin (TOR) plays an important regulatory role in both animal and plant regeneration. In plants, TOR is involved in the regulation of roots, stem growth, and callus formation [22,23,24], and in animals, TOR is a central hub for integrating nutrients, energy, hormones, and environmental signals [25,26]. Cell growth and cell cycle progression are generally tightly connected, allowing cells to proliferate continuously while maintaining their size. TOR is an evolutionarily conserved kinase that regulates both cell growth and cell cycle progression coordinately [27]. Stem cells and their metabolites have great application value in agriculture and regenerative medicine. The advancement in regenerative medicine benefits human health, and it has great prospects in the medical field [28]. Stem cells can be regarded as ideal seed cells for genetic engineering, able to the repair damaged tissues and organs and to overcome immune rejection. In this review, we discuss the regeneration mechanisms of animals and plants, highlighting the similarities and differences between these biological processes. Additionally, we summarize the main recent findings on animal and plant stem cells in the field of regeneration, and provide new ideas and directions for the protection of endangered species and the development of regenerative medicine.

## 2. Similarities and Differences in Plant and Animal Regeneration

Plants have the remarkable ability to drive cellular dedifferentiation and regeneration [19]. However, the regenerative capacity of animals varies greatly across different species. Invertebrates and amphibians generally have a high regenerative capacity [29]. In contrast, the regeneration capacity of vertebrates, such as mice, is relatively weak [30,31,32]. Whether the research subject is a planarian with strong regenerative capacity or a human with weak regeneration capacity, the fundamental mechanism of regeneration is the differentiation of stem cells into the damaged/missing tissues.

The regeneration processes of animals and plants have certain similarities. Firstly, they can be divided into the same levels of regeneration, including cell, tissue, structural, organ, and systemic regeneration [33]. Secondly, in both plants and animals, injury is the main stimulus for the formation of specialized wound tissue that initiates regeneration. A regenerative response from these organisms can be elicited by environmental insults, such as pathogens or even predatory attacks. Amputation in animals is usually, but not always, followed by the formation of a specialized structure known as a regeneration blastema. This structure consists of an outer epithelial layer that covers mesodermally derived cells, inducing a canonical epithelial/mesenchymal interaction, a conserved tissue relationship central to the development of complex structures in animals [34]. In plants, a frequent, but not universal, feature of regeneration is the formation of a callus, a mass of growing cells that has lost the differentiated characteristics of the tissue from which it arose. A callus is typically a disorganized growth, arising on wound stumps and in response to certain pathogens. One common mode of regeneration is the appearance of new meristems within callus tissue. Therefore, the plant callus and animal blastema share the characteristics of being specialized yet undifferentiated structures capable of regenerating new tissues [4]. Moreover, the process of stem cell regeneration induced by somatic cells in plants is similar to that induced by animal pluripotent stem cells. In animals, the production of induced pluripotent stem cells (iPSC) depends on the expression of many key transcription factors. Similar to animal cells, the induction and maintenance of stem cells in plants also depend on the induction and expression of several key transcription factors, such as class B-ARR, WUSCHEL (WUS), or WUSCHEL RELATED HOMEOBOX5 (WOX5). Therefore, the stem cells induced in plants that express the pluripotent genes such as *WUS* or *WOX5* can also be called plant iPSC [35]. In addition, the regeneration of animals and plants requires the participation of stem cells.

The regenerative capacity of animals and plants varies greatly. Generally speaking, the regenerative capacity is weak in higher animals, and varies greatly between body parts (Figure 1). The skin, as well as other microorgans and tissues of animals, have relatively fast renewal speeds and strong regeneration capacities [36]. The regeneration capacity of the heart, stomach, and other organs is weak, whereas that of the liver is relatively strong [37]. Unlike certain nerve tissues that still retain axonic connections, animal nerve cells have almost no regenerative capacity; therefore, certain types of brain cell damage and senile dementia are irreversible and can only be repaired via stem cell treatment [38]. The regenerative ability of plants is generally stronger than that of animals, but also vary greatly between species. For example, the regenerative capacities of *Taxus chinensis, Metasequoia glyptostroboides*, and *Ginkgo biloba* are relatively weak, whereas those of lower plants, such as *Ficus virens, Laminaria japonica*, and *Undaria pinnatifida*, are relatively strong [39].

Stem cells are divided into totipotent stem cells, pluripotent stem cells, and unipotent stem cells [40]. The distribution of animal and plant stem cells is also quite different. In plants, stem cells existing in the shoot apical meristem (SAM) and root apical meristem (RAM) are pluripotent, and plant stem cells mainly exist in the meristem of plants for a long time [41]. Meristems can differentiate into vegetative tissues, protective tissues, conducting tissues, mechanical tissues, secretory tissues, and other plant cell populations with identical physiological functions and morphological structures to form vegetative and reproductive organs of plants [42,43]. In addition, plants can also produce calluses, which are similar to stem cells, and are the tissue formed by somatic cells in response to injury and dedifferentiation [19,44,45]. There is often a lack of stem cell aggregation in animal tissues; however, they are widely distributed in various tissues and organs, though in small numbers [46]. In addition, due to the differences in evolution, there are significant differences in the signal pathways and regulators regulating plant and animal regeneration (Table 1 and Table 2). In plants, a feedback regulation pathway is formed between WUS and CLAVATA (CLV), which regulates the steady state of stem cells in stem tips [47]. The SHORTROOT (SHR)-SCARECROW (SCR) signaling pathway plays a key role in maintaining apical meristems [48,49]. In animals, the Wnt and Notch classical signaling pathways regulate self-renewal of hematopoietic, intestinal epithelial, skin, and neural stem cells [50].

## 3. Molecular Mechanisms of Plant and Animal Regeneration

There are great differences in the regenerative capacities of animals and plants, and the involved signaling pathways are also different. Even in plants, the transcription factors and signal pathways regulating SAM and RAM regeneration vary [80]. SAM is formed in the early stage of embryonic development and is structurally divided into the central zone (CZ), rib zone (RZ), and peripheral zone (PZ). The CZ region is composed of pluripotent stem cells in an undifferentiated state, with a long cell division cycle; the RZ region provides cell support for the vascular meristem; and the PZ region is the core region for further cell division, differentiation, and development into lateral organs [81]. In SAM, STM and WUS are essential for stem cells to remain undifferentiated [57]. STM can inhibit the differentiation while maintaining the proliferation of meristem cells, and can also integrate mechanical signals that play a role in the formation of lateral organs [82]. Plant stem cells require induction niches. In SAM, this role is played by cells located in the organizing center (OC). At the molecular level, the OC is defined by highly localized expression of the homeodomain transcription factor WUS [83]. WUS fluidity is highly directional, but its specific mechanism has not yet been elucidated. CLV3, as a major stem cell-derived signal, connects WUS with STM. In *Arabidopsis*, WUS and STM form heterodimers and combine with the promoter region of *CLV3*, ensuring a stable number of stem cells [56]. CLV3 is a short secretory peptide modified after processing and translation. CLV3 peptides diffuse in the interstitial space and act by binding with a group of related leucine rich repeat (LRR) receptor complexes found on the plasma membrane [84]. The joint action of these receptors is to combine with CLV3 to activate intracellular signaling cascades. The net effect of CLV signaling is reduced WUS expression, defining a local negative feedback loop to induce WUS migration from the OC to stem cells in order to maintain their fate [85]. In addition, *STM* gene expression depends on *WUS*, and *WUS*-activated *STM* expression enhances WUS-mediated stem cell activity (Figure 1) [47,56].

In addition, in SAM, the local regulatory system appears insufficient to synchronize stem cell behavior without developmental or environmental input. Communication between peripheral developmental organs and central stem cells in SAM is mainly controlled by phytohormones, among which auxin and cytokinin have the greatest impact [86]. Cytokinin acts as a cell cycle inducer and is important for WUS activation, while auxin mainly triggers peripheral differentiation [87]. Interestingly, auxin also enhances the output of cell division proteins by directly inhibiting the expression of negative feedback regulators of cytokinin signal transduction [86]. Recent studies have found that TOR kinases play a central role in metabolism, light-dependent activation of *WUS*, and stem cell activation in SAM [23]. RAM is mainly regulated by the auxin-dependent PLT pathway and the auxin-independent SHR/SCR pathway [88,89]. Key transcription factors such as *SHR*, *SCR,* and *PLT1/2/3/4* play a crucial role in the organization and maintenance of RAM. SCR is expressed in the static center and endothelium, and SHR is expressed in the periapical stele cells. Both are necessary to maintain static center function and jointly provide signals for the stem cell microenvironment [48,49]. In addition, PLTs strongly affect the characteristics, cell expansion, and differentiation of stem cells and RAM by forming gradients which depend on the stability and movement of PLT proteins [90,91]. PLTs and auxin gradients are correlated, but also partially independent (Figure 1A) [92,93].

The regeneration process includes tissue repair, de novo organ regeneration, the formation of wound-induced calluses, and somatic embryogenesis. Root tip repair involves a wounding response, redistribution of auxin and cytokinin, reconstruction of the quiescent center (QC), and stem cell niche re-establishment [94]. Studies have found that damage-induced jasmonic acid (JA) signaling can also activate stem cells to promote regeneration, and JA signaling regulates the expression of the RETINOBLASTOMA-RELATED (RBR)-SCR molecular network and stress response gene *ERF115* to activate the root stem cell tissue center, thereby promoting root regeneration. Auxin activates *WUSCHEL RELATED HOMEOBOX11/12* (*WOX11/12*) to transform root-initiating cells into the root primordium. During this process, the expression level of *WOX11/12* decreases, whereas that of *WOX5/7* increases. The WOX11/12 protein directly binds to the WOX5/7 promoter to activate its transcription, whereas WOX5/7 mutation leads to defects in primordium formation [65]. At the genetic level, the highly specific and QC-expressed gene *WOX5* delineates QC identity and maintenance [95]. WOX5 activity most likely occurs through direct effect on cell cycle regulators. Plants with disrupted expression levels of WOX5 show aberrant differentiation rates of the distal stem cells, indicating the role of WOX5 in preventing stem cell differentiation [96]. In contrast to SAM, where auxin triggers differentiation, hormones need to specify niches and maintain cell proliferation in RAM. Cytokinin mainly acts far away from the root tip and promotes differentiation through mutual inhibition with auxin [97]. However, cytokinins have also been shown to counteract the unique properties of QC cells by reducing auxin input from the surrounding environment and inducing cell division [98]. Maintaining stem cell homeostasis in the stem and root niches is essential to ensure that sufficient numbers of new cells are generated to replace removed cells, as well as the proper differentiation and growth and formation of new tissues and organs. It is worth noting that RBR protein is a plant homologue of RB (a tumor suppressor protein) and plays a crucial role in SAM and RAM [99,100]. Like in animals, RBR in plants inhibits cell cycle progression by interacting with E2F transcription factor homologues. In addition, decreased RBR levels lead to increased numbers of stem cells, while increased RBR levels lead to stem cell differentiation, indicating that RBR plays an important role in stem cell maintenance. At present, RBR is a protein known to be involved in stem cell function, and is conserved between the animal and plant kingdoms [1]. Interestingly, TOR not only plays a role in SAM stem cell activation, but also promotes QC cell division in RAM (Figure 1A) [101].

De novo root regeneration is the process by which adventitious roots form from wounded or detached plant organs. Auxin is the key hormone that controls root organogenesis, and it activates many key genes involved in cell fate transition during root primordium establishment [102]. The detached leaves of Arabidopsis thaliana can regenerate adventitious roots on hormone-free medium [103]. From 10 min to 2 h after leaf detachment, a wave of JA is rapidly produced in detached leaves in response to wounding, but this wave disappears by 4 h after wounding [104]. JA activates the expression of transcription factor gene *ERF109* through its signaling pathway, which, in turn, up-regulates the expression of *ANTHRANILATE SYNTHASE α1* (*ASA1*). ASA1 is involved in the biosynthesis of tryptophan, a precursor of auxin production. After 2 h, the concentration of JA decreased, resulting in the accumulation of JAZ protein, which could directly interact with ERF109 and inhibit ERF109, thus turning off the wound signal. In general, the post-injury JA peak promotes auxin production and, thus, promotes root regeneration from the cuttings. Root organogenesis also requires a strict turning-off of the JA signal [105].

Callus formation is one of the most important methods of plant regeneration. Studies have analyzed why calluses have regenerative capacity. Through single cell sequencing of *Arabidopsis* hypocotyl calluses, researchers confirmed that calluses are similar to the root primordium or root tip meristem, and can be roughly divided into three layers: the outer cells are similar to the epidermis and root cap of the root tip, the middle layer cells to the quiescent center (QC), and the inner cells to root tip initial vascular cells. It was found that middle layer cells of calluses had highly similar transcriptome characteristics to the QCs of root tip resting centers, and were also source stem cells for root and bud regeneration [59]. AAR12, of the cytokinin signal transduction pathway, is the main enhancer of callus formation [62]. *APETALA2/ETHYLENE RESPONSE FACTOR* (*AP2/ERF*) transcription factors, such as *WIND1*, *ERF113/RELATED TO AP2 L* (*RAP2.6L*), *ESR1*, and *ERF115,* in *Arabidopsis thaliana* are key regulators of rapid post-traumatic-induced regeneration when wounded. Wounding upregulates cytokinin biosynthesis and signal transduction, thereby promoting cell proliferation and callus formation [60,106,107,108]. WIND1 can promote callus formation and shoot regeneration by upregulating ESR1 (Figure 1A) [45].

Plants can undergo multiple regenerative processes after wounding to repair wounded tissues, form new organs, and produce somatic embryos [109]. Plant somatic embryogenesis refers to the process by which somatic cells produce embryoids through in vitro culture [110]. This process can occur directly from the epidermis, sub-epidermis, cells in suspension, protoplasts of explants, or from the outside or inside of a callus formed from dedifferentiated explants. The transformation from somatic cells to embryogenic cells is the premise of somatic embryogenesis. In this process, the isolated plant cells undergo dedifferentiation to form a callus. The callus and cells undergo redifferentiation into different types of cells, tissues, and organs, and finally generate complete plants [111]. This process involves cell reprogramming, cell differentiation, and organ development, and is regulated by several transcription factors and hormones [112]. For example, the *WUS* gene regulates the transformation of auxin-dependent vegetative tissues to embryonic tissues during somatic embryogenesis [113,114]. Overexpression of WUS can induce somatic embryogenesis and shoot and root organogenesis. Ectopic expression of the WUS gene can dedifferentiate recalcitrant materials that do not undergo somatic embryogenesis easily to produce adventitious buds and somatic embryos [115]. Additionally, *LEAFY COTYLEDON 1* (*LEC1*), highly expressed in embryogenic cells, somatic embryos, and immature seeds, can promote somatic cell development into embryogenic cells. Furthermore, *LEC1* can maintain the fate of embryogenic cells at the early stage of somatic embryogenesis. At present, *LEC1* is used as a marker gene for somatic embryogenesis in several species [116]. Unlike LEC1, *LEC2* can directly induce the formation of somatic embryos, which may activate different regulatory pathways [117].

In recent years, through research on animals with strong regeneration capacities, such as planarians, leeches, and salamanders, it was found that the early stages of regeneration are jointly regulated by cell death/apoptosis-related genes, MAPK signal-related genes, and *EGR* [118]. In plants, programmed cell death (PCD) plays crucial roles in vegetative and reproductive development (dPCD), as well as in the response to environmental stresses (ePCD) [119,120]. Sexual reproduction in plants is important for population survival and for increasing genetic diversity. During gametophyte formation, fertilization, and seed development, there are numerous instances of developmentally regulated cell elimination, several of which are forms of dPCD essential for successful plant reproduction [121]. In the late stages of regeneration, many signal pathways participate in cell proliferation and regulation of various responses. The Wnt signaling pathway is widely distributed in invertebrates and vertebrates, and is a highly conserved pathway during evolution. Wnt signaling plays an important role in early embryonic development, organ formation, tissue regeneration, and other physiological processes [69,122,123]. Wnt proteins are a family of 19 highly conserved secretory glycoproteins that act as ligands for several receptor-mediated signaling pathways, including those that regulate processes throughout development [123]. The classic Wnt signaling pathway is mainly mediated by β-catenin. β-catenin is a multifunctional protein which helps cells respond to extracellular signals and influences by interacting with the cytoskeleton [124]. When Wnt binds to its membrane receptor, Frizzled (FZD), it activates the intracellular protein Dvl. Dvl receives upstream signals in the cytoplasm and is the core regulator of the Wnt signaling pathway. Wnt inhibits the function of the β-catenin degradation complex formed by APC, AXIN, CK1, glycogen synthase kinase 3β (GSK3β), and other proteins, thus stabilizing β-catenin in the cytoplasm. Stably accumulated β-catenin in the cytoplasm enters the nucleus and binds to the TCF/LEF transcription factor family to initiate the transcription of downstream target genes, such as *c-Myc* and *cyclin D1*, in order to promote regeneration. TCF/LEF transcription factor’s association with β-catenin initiates the expression of key genes in the multiple Wnt signaling pathways [50,125]. The Wnt signaling pathway is important for human development and the maintenance and regulation of adult stem cells, but improper Wnt activation can lead to carcinogenesis [126]. For example, in the differentiation of mouse embryonic stem cells (mESC), Wnt activation of β-catenin signaling inhibits myocardial differentiation and promotes endothelial and hematopoietic lineage differentiation. During vertebrate embryonic development, Wnt activation induces ESCs to enter the anterior and posterior lamellar mesoderm (LPM). In pre-LPM, Dickkopf (Dkk) is secreted from the endoderm, preventing Wnt from binding to its receptor and leading to the induction of the cardiogenic mesoderm and the formation of cardiac progenitor cells (CPC) (Figure 1B) [123].

Similar to Wnt signaling, Notch signaling is a highly conserved signaling pathway that is widely involved in various regeneration processes in different organs, such as the tail fin, liver, retina, spinal cord, and brain [127]. Notch signaling also plays an important role in the self-renewal and differentiation regulation of stem cells. In stem cell biology, Notch signal transduction is highly environmentally dependent, and the biological consequences of pathway activation vary from maintaining or expanding stem cells to promoting stem cell differentiation [128]. Researchers found that Notch receptors and ligand expression were up-regulated during zebrafish fin regeneration in 2003, and many studies have also shown that Notch signaling plays a key role in fin repair, regulating venous arterialization, and cell proliferation and differentiation [129]. Notch signaling can also regulate duct cell accumulation and biliary tract differentiation, promote the expansion and differentiation of liver progenitor cells, and antagonize Wnt signaling during liver regeneration. However, different Notch receptors have different effects on hepatocytes, confirming the complex functions of Notch signaling in the treatment of liver diseases [130]. Notch signaling is mediated by the interaction between Notch ligands and receptors in adjacent cells. There are four kinds of Notch receptors (Notch1-4) in mammals, which are composed of three parts: the extracellular domain (NEC), transmembrane domain (TM), and intracellular domain (NICD). The Notch protein is cleaved three times, and its NICD is released into the cytoplasm and enters the nucleus to bind to the transcription factor CBF-1, suppressor of hairless, Lag (CSL) to form a transcriptional activation complex. The CSL protein is a key transcriptional regulator in the Notch signaling pathway, which is also known as the classical Notch signaling pathway or the CSL-dependent pathway. It activates the Hairy Enhancer of Split (HES), Hairy, and Enhancer of split-related genes with the YRPW motif (HEY), homocysteine-induced ER protein, and other basic helix–loop–helix (bHLH) transcription factor families of the target genes [131,132]. For example, Notch signaling can enhance bone regeneration in the mandibles of zebrafish, and is reactivated after valvular damage in zebrafish larvae and adults, which is necessary in the initial stage of heart valve regeneration (Figure 1B) [133].

In addition, the more conserved Hh pathway also plays a key role in adult tissue maintenance, renewal, and regeneration [134]. The Hh protein has been identified in many animals, from jellyfish to humans. Drosophila has only one *Hh* gene, while vertebrates have 3–5. All Hh proteins are composed of the N-terminal “Hedge” domain and the C-terminal “Hog” domain. The Hedge domain mediates protein signaling activity. The Hog domain can be further subdivided into the N-terminal Hint domain and the C-terminal sterol recognition region (SRR). The N-terminal Hint domain is sequentially similar to the self-splicing intron, and the C-terminal SRR binds to cholesterol [135]. Hh signal transmission is mediated by two receptors on the target cell membrane, Patched (Ptc) and Smoothened (Smo). The receptor Smo is encoded by the proto-oncogene Smoothened and is homologous to the G-protein-coupled receptor. It is composed of a single peptide chain with seven transmembrane regions. The N-terminal is located outside the cell, and the C-terminal is located inside the cell. The amino acid sequence of the transmembrane region is highly conserved [136]. The serine and threonine residues at the C-terminal are phosphorylated sites. When protein kinase catalyzes, it binds phosphate groups. The members of this protein family have the function of a transcription promoter only when they maintain their full length and start the transcription of downstream target genes. When the carboxyl end is hydrolyzed by the proteasome, a transcription inhibitor is formed to inhibit the transcription of downstream target genes. Smo is a necessary receptor for Hh signal transmission. Glioma-associated oncogene transcription factors (GLI) are transcriptional effectors of the Hh pathway. Stimulated by Hh signal transduction activation, GLI proteins are differentially phosphorylated and processed into transcriptional activators that induce the expression of Hh target genes to initiate a series of cellular responses, such as cell survival and proliferation, cell fate specification, and cell differentiation [137,138]. A previous study found that Hh signaling mediates liver regeneration by regulating DNA replication and cell division. Treatment of mice with Hh inhibitors caused a slowing of cell proliferation and mitotic arrest, which led to the inhibition of liver regeneration. Mice treated with the Hh inhibitor vismodegib showed inhibited liver regeneration, accompanied by significant decreases in the expression of Hh-inducible factors GLI1 and GLI2 (Figure 1B) [139].

The Hippo signaling pathway is a major regulator of cell proliferation, tissue regeneration, and organ size control [132]. Hippo is highly conserved in mammals, controlling development and tissue organ homeostasis; imbalances can lead to human diseases such as cancer [140]. The core of the Hippo pathway is the kinase cascade; that is, mammalian STE20-like1/2 (Mst1/2) (Hippo homolog) and Salvador 1 protein (SAV1) form a complex that phosphorylates and activates large tumor-suppressing kinases (LATS1/2). LATS1/2 phosphorylates and inhibits transcription coactivators such as Yes-associated proteins (YAP) and transcriptional coactivators with PDZ-binding motifs (TAZ) [141]. LATS1/2 is a protein kinase that plays an important role in the Hippo signaling pathway, and exhibits anticarcinogenic activity. LATS1/2 deletion enhances TAZ/YAP activity and directly activates oncogene expression [142]. During tissue damage, the activity of YAP, the main effector of the Hippo pathway, is instantaneously induced, which in turn promotes the expansion of tissue-resident progenitor cells and promotes tissue regeneration [143]. Recent animal model studies have shown that the induction of endogenous cardiomyocyte proliferation is crucial for cardiac regeneration, and inhibition of Hippo signaling can stimulate cardiomyocyte proliferation and cardiac regeneration [144]. TGF-β superfamily signal transduction plays an important role in regulating cell growth, differentiation, and development in many biological systems [145]. TGF-β signaling phosphorylates Smad proteins and transports them to the nucleus. Activated Smad proteins regulate a variety of biological processes by binding to transcription factors, leading to cell state-specific transcriptional regulation [146]. For example, TGF-β signaling in zebrafish promotes cardiac valve regeneration by enhancing progenitor cell proliferation and valve cell differentiation. In addition, TGF-β superfamily members also play important roles in the steady renewal and regeneration of the adult intestine (Figure 1B) [147,148].

TOR signaling pathways are present in both animals and plants, and are also associated with regeneration. Plant growth is affected by light and glucose, which are known activators of the TOR pathway [149]. The TOR signaling pathway is involved in root and stem growth and callus formation, and TOR phosphorylates downstream cell cycle factor E2Fa to promote these processes [22,24]. Moderate expansion of the Akt gene in animals activates the mTOR signaling pathway and promotes cell proliferation [150]. GSK3β is a direct substrate of Akt and is inhibited by Akt during animal regeneration [151]. *BR-INSENSITIVE 2* (*BIN2*) was the first plant GSK3-like kinase to be characterized by genetic screening. The kinase domain of the GSK3-like kinase found in *Arabidopsis* and rice has 65–72% sequence homology to human GSK3β [25,152]. Biochemical and genetic analyses have confirmed that BIN2 plays a negative role in BR signal transduction and the regulation of cell growth. However, in plants, it was found that TOR can regulate the phosphorylation level of the neglected ribosomal protein S6 kinase beta 2 (S6K2), and S6K2 can interact with BIN2 to directly phosphorylate BIN2 and regulate plant growth [153]. The conserved characteristics of TOR signaling in the normal physiology and regeneration of animals and plants suggest its important role in maintaining normal physiological homeostasis of animals and plants.

## 4. Applications of Regeneration Technology

The growth and development of animals and plants is a process of differentiation from pluripotent stem cells (fertilized eggs) to pluripotent stem cells, and then to specialized stem cells [154]. On the contrary, the terminally differentiated cells of animals and plants carrying complete genetic material also have the potential to transform into stem cells. In plants, somatic cells can restore their totipotency through dedifferentiation and regenerate intact plants. Consistent, in the study of animal cell dryness, it was also found that four transcription factors, octamer binding transfer factor 4 (Oct4), SRY box transfer factor 2 (Sox2), Kruppel like factor 4 (Klf4), and c-Myc, were transferred into mouse fibroblast cells, which can cause them to become iPSC. This discovery indicates that immature cells can develop into all types of cells [155,156]. Stem cells and their metabolites, from both plants and animals, are widely used in agriculture, animal husbandry, and regenerative medicine (Figure 2).

Plant totipotent stem cells have good application potential in crop breeding. The totipotent stem cells of animals in the placenta can be cryopreserved to treat some diseases after adulthood. Plant pluripotent stem cells and their metabolites can be used in the development of drugs, health foods, and cosmetics. For animals, iPSC can produce various necessary organs, but at present, due to ethical constraints, artificial organs have not been allowed [154]. Artificial meat that can be made from animal multipotent stem cells can also be used for pet disease treatment. The unipotent stem cells of plants are also used for the extraction of some pigment substances. In addition, purple shirt stem cells in a suspension culture can produce anti-cancer substances such as Taxamairin A and B, and the unipotent stem cells in milk have therapeutic potential in treating some animal diseases [157].

In agriculture, plant genetic transformation and callus culture are key processes in crop gene editing and breeding [158]. A previous study found that overexpression of the wheat WUSCHEL family gene TaWOX5 can significantly improve transformation efficiency, and that callus culture can aid wheat transgenics [159]. In *Arabidopsis*, the injury-inducing factor WIND1 can promote callus formation and bud regeneration by upregulating *Arabidopsis* ESR1 expression, and the *esr1* mutant shows defects in callus formation and bud regeneration [61]. This finding is of great significance for in vitro plant tissue culture. Regenerating adventitious roots from cuttings is a common plant clonal reproduction biotechnology in the forestry and horticulture industries. Plant somatic embryogenesis also has broad application prospects in artificial seeds, haploid breeding, asexual reproduction, and germplasm conservation [160]. Plant viral diseases are serious agricultural diseases, significantly affecting the yield/quality of crops and leading to crop failure. Stem tip virus-free technology is the only effective biotechnology to be found thus far that can remove viruses from plants. It has been widely used in agricultural production to obtain virus-free seedlings, and has also been applied in potatoes, fruit trees, flowers, and other crops. Stem cells and their daughter cells of SAM from *Arabidopsis thaliana* can inhibit infection with the cucumber mosaic virus (CMV). The mechanism study found that viruses cause local WUS protein induction and accumulation in stem cells, as well as subsequent migration to surrounding compartments. By directly inhibiting protein synthesis in cells, the replication and transmission of viruses can be restricted, which can protect stem cells and their differentiated daughter cells from viral infections [161]. The WUS protein has anti-viral characteristics in plant stem cells, and can help plants resist viral invasion.

With growth in the global population and meat demand, the harmful effects of animal husbandry on the environment and climate will increase [162]. Moreover, animal-borne diseases and antibiotic resistance are harmful to humans [163]. A suggested method to reduce the consumption of animal meat is to increase the production of artificial meat through species iPSCs, which can also eliminate many environmental and ethical issues which occur with traditional meat production [164]. In 2013, Dutch biologist Mark Post produced the first piece of artificial meat in history by using the animal cell tissue culture method, which attracted widespread attention [165]. Animal cell culture artificial meat is mainly composed of skeletal muscle containing different cells. These skeletal muscle fibers are formed by the proliferation, differentiation, and fusion of embryonic stem cells or muscle satellite cells. They first isolated the growth-differentiable primitive stem cells. By adding a culture medium rich in amino acids, lipids, and vitamins, they accelerated cell proliferation and differentiation and obtained a large number of bovine muscle tissue cells [166]. The production of cultured meat requires robust cell sources and types. In order to achieve the scale required for the commercial production and sales of cultured meat products, it is necessary to further develop immortal special cell lines. In addition to technical challenges, the relationship between cultured meat and social/cultural phenomena and social systems must also be considered [167]. In the racing industry, tendon and ligament injuries are common problems that can end the careers of racehorses. Therefore, stem cell therapy has received attention in this field. Common clinical applications include the use of stem cells to treat tendon and ligament strains in the joints of horses [168].

Stem cell technology also has applications in the medical beauty industry. Some plants contain raw materials needed in cosmetics, and stem cell culture can overcome barriers such as low endogenous content and difficult extraction methods [169]. For example, plant cell culture technology can be used to derive certain mint-based hair care products [170,171]. Plants containing antioxidant substances, such as grapes and cloves, can be used in anti-ultraviolet light protection skincare products. Plant stem cells can be used to obtain these antioxidant components at a more efficient rate [172]. Although plant stem cells are widely used in the medical beauty field, their full potential remains to be explored due to the lack of scientific evidence and the large variety of flora that may have potential for stem cell culture. In addition, *Taxus chinensis* and *Catharanthus* roseus suspension cell cultures can also be used to produce taxol- and vinblastine-based anticancer substances [173,174]. Although promising advances have been made in the field of plant stem cells and their various applications, it is unclear whether plant-derived extracts and stem cell extracts have race-specific effects in humans.

Regenerative medicine is a new research area in the field of medicine. It uses biological and engineering methods to create lost or damaged tissues and organs so that they mimic the structure and function of normal tissues and organs [175]. At present, stem cell therapy is a widely used type of regenerative medicine therapy, and plays an important role in the treatment of chronic diseases, including autoimmune diseases, leukemia, heart disease, and urinary system problems [28,176]. Autoimmune Addison’s disease (AAD) is an inevitably fatal disease in the absence of treatment. Affected patients must receive steroid replacement for life to survive. Studies have found that AAD can be improved by manipulating endogenous adrenal cortical stem cells to enhance adrenal steroid production [177]. Hematopoietic stem cell transplantation can be used to treat leukemia, and around 80–90% of leukemia patients show improvement after hematopoietic stem cell transplantation, of which 60–70% enter remission [178]. The cardiac regenerative medicine field is currently facing challenges due to the lack of cardiac stem cells in adults, low turnover rate of mature myocardial cells, and difficulty in providing treatment for injured hearts. At present, cell reprogramming technology has been applied to generate patient-specific myocardial cells through both direct and indirect methods [179]. Stem cell therapy can also be used to treat stress-induced urinary incontinence, and preclinical studies have made advances in regenerating the urethral sphincter by using secretory group cells or chemokines that can return repair cells to the injured site [180].

In addition, regenerative medicine is closely related to tissue engineering. At present, organ transplantation is still widely used to replace failed tissues and organs. However, with substantial increases in the demand for organ transplantation in recent decades, it is difficult to maintain an adequate supply of available organs [181]. The emergence of 3D biological printing technology has made up for the lack of supply of tissues and organs. Compared with traditional tissue engineering methods, 3D bioprinting utilizes a more automatic process and can create more advanced scaffolds with accurate anatomical characteristics, allowing the precise co-deposition of cells and biomaterials [182]. 3D biological printing technology is also used in cancer research, drug development, and even clinician/patient education [183]. However, there are still some issues with 3D biological printing technology, such as limitations with biological inks and printers, as well as the size of the end product. At present, bioprinted tissues are often small and composed of only a few cell types, resulting in limited function and scalability [184,185]. In addition, the cost of 3D biological printing is high, and the resolution requires further improvement.

Although stem cell therapy has good outcomes, it also has safety risks. For example, pluripotent stem cells have the ability to form teratomas themselves [186]. The IPS cells established using retroviral vectors are used to introduce exogenous genes, and their expression may be retained or reactivated during differentiation. This may have impacts on the directivity and carcinogenicity of differentiation [187]. To fully realize the benefits of regenerative medicine, the real and imaginary boundaries of social, ethical, political, and religious views must be addressed [188,189]. We must carefully measure the potential therapeutic benefits of the clinical application of stem cells and weigh them according to the possible side effects in each patient and disease indication, because the clinical use of stem cells can lead to overly high expectations. Our decision-making process regarding disease management should continue to firmly follow the conservative principles of evidence-based medicine.

## 5. Conclusions and Future Perspectives

At present, it is not uncommon to utilize stem cells in both medicine and agriculture, such as for the effective repair of damaged tissues and organs and to treat cardiovascular and metabolic ailments, as well as diseases of the nervous system, blood system, and others [161,190,191]. Recently, some research has revealed the “switch” mechanism underlying the brain regeneration of salamanders, and constructed a space–time map of brain development and regeneration of single salamander cells [192]. The next step in this field is to achieve brain regeneration in mammals, including humans, which would involve the activation of brain “seed cells” and the introduction of key factors, thus turning on the “switch” of human brain regeneration. It is expected that new treatment methods will soon be developed to improve the clinical rehabilitation of patients with brain diseases. In addition, the potential value of stem cells in anti-viral applications is of great interest. Plant stem cells can resist viruses, and animal stem cells can also use antiviral Dicer (AviD) to resist the invasion of multiple RNA viruses [193]. The antiviral mechanism of stem cells may be of great value for future medical and pharmaceutical research on human viral infection resistance.

Plant regeneration is mainly carried out through somatic embryogenesis or organogenesis; however, plant regeneration can be promoted by transferring plant-related genes [194]. With the rapid development of synthetic biology, this concept has been applied to the regeneration of animals and plants [195]. The concept of “build-to-understand” synthetic biology is instructive in the field of tissue regeneration, where more extensive and flexible research can be achieved by building genetic circuits. Using synthetic biology, we can import genes with strong regenerative abilities into rare and precious plants to increase their yield. CRISPR-Cas9 technology enables genome-wide epigenetic modifications to modify plant regeneration pathways or affect specific gene loci to regulate plant regeneration [196,197,198].

In this review, we have made a more detailed and systematic summary of the research of animal and plant stem cells in the field of regeneration in recent years, and described the regeneration mechanisms of animals and plants. In addition, we also proposed the application prospects of stem cells in agriculture, animal husbandry, and regenerative medicine, which would provide new ideas and directions for the protection of endangered species and the development of regenerative medicine. However, due to the lack of existing genetic information on higher animals and plants, current research is mainly focused on simpler species, such as *Arabidopsis*, planarians, etc. There is still a long way to go before applications in advanced endangered plants and regenerative medicine can be fully realized. However, with rapid developments in synthetic biology, single cell sequencing, and other technologies, research on higher animals and plants is becoming more feasible. It is believed that with more research, the mystery of regeneration will eventually be solved.

## Figures and Tables

**Figure 1 ijms-24-04392-f001:**
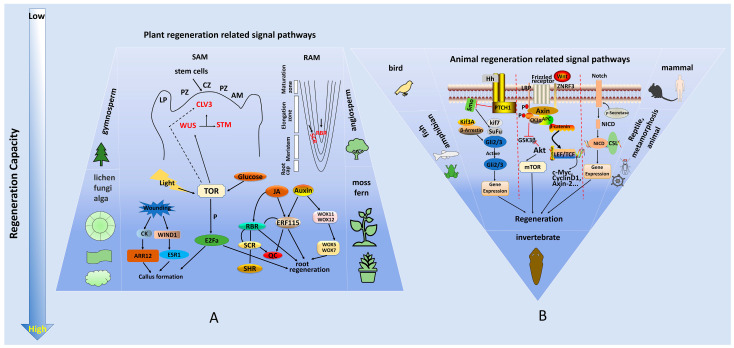
Comparison of the regenerative capacities and mechanisms of animals and plants. (**A**) Signal pathways related to plant regeneration. (**B**) Signal pathways related to animal regeneration. The regenerative capacities of different plants have slight differences, and these differences are expressed by trapezoid. The regeneration abilities of different animals vary greatly, so the regeneration differences are represented by a triangle. SAM: shoot apical meristem; CZ: central zone; RZ: rib zone; PZ: peripheral zone; AM: axillary meristem; RAM: root apical meristem; QC: quiescent center. RAM is composed of the maturation zone, elongation zone, meristem zone, and root cap.

**Figure 2 ijms-24-04392-f002:**
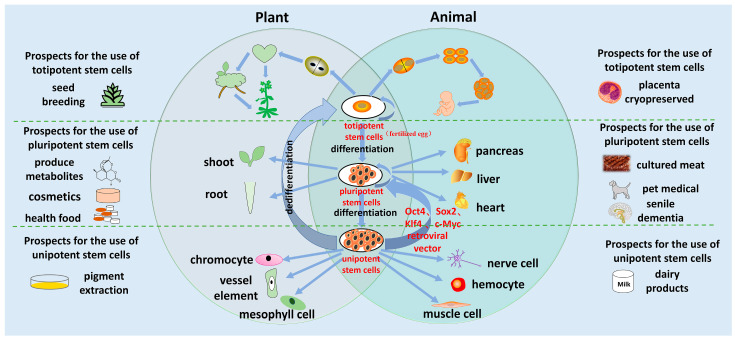
Regeneration of animals and plants, and application of their stem cells.

**Table 1 ijms-24-04392-t001:** Summary of plant regeneration-related genes.

Regeneration-Related Genes	Related Signaling Pathways	Promote or Inhibit Regeneration	Functions	References
*WUS*	Auxin signal, CLV3-WUS feedback pathway	promote	Stem cell maintenance (SAM)	[51,52,53]
*CLV3*	Cytokinin signal, CLV3-WUS feedback pathway	inhibit	Maintains stem cell balance (SAM)	[51,54,55]
*SHOOTMERISTEMLESS* (*STM*)	Cytokinin signal	promote	Establishment and maintenance of stem end meristem	[56,57]
*PLETHORAs (PLTs*)	JA signal, PLT pathway	promote	Maintains the niche of root stem cells	[58,59]
*WOUND INDUCED DEDIFFERENTIATION 1* (*WIND1*)	Cytokinin signal	promote	Promotes callus formation	[60,61]
*Type-A ARRs* (*RRAs*)	Cytokinin signal	inhibit	Inhibits callus formation and bud regeneration	[62]
*Type-B ARRs* (*RRBs*)	Cytokinin signal	promote	Promotes callus formation and bud regeneration	[62]
*ENHANCER OF SHOOT REGENERATION 1* (*ESR1*)	Auxin signal Cytokinin signal	promote	Promotes callus formation and bud regeneration	[61,63]
*MYC2*	JA signal	promote	Promotes callus formation	[64]
*LATERAL ORGAN BOUNDARY DOMAINs* (*LBDs*)	Auxin signal	promote	Promotes callus formation	[63]
*WOX5/7*	Auxin signal Cytokinin signal	promote	Stem cell maintenance (RAM)	[65]
*SHR/SCR*	Auxin signal	promote	Maintains the niche of root stem cells	[49]
*ETHYLENE RESPONSE FACTOR 115 (ERF115*)	JA signal	promote	Activates the root stem cell and promotes regeneration	[64]
*ERF109*	JA signal	promote	Activates the root stem cell and promotes regeneration	[64]
*Auxin Response Factors* (*ARFs*)	Auxin signal	promote	Regulates development and shoot regeneration	[66,67]

**Table 2 ijms-24-04392-t002:** Summary of animal regeneration-related genes.

Regeneration-Related Genes	Related Signaling Pathways	Promote or Inhibit Regeneration	Functions	References
*Wnt1*	Wnt signaling	promote	Promotes regeneration of organs and tissues	[68,69]
*P21*	Notch signaling	inhibit	Inhibits the regeneration of the liver and other organs	[70]
*Prod1* (*Salamander*)	unsure	promote	Promotes salamander limb regeneration	[71]
*G protein nucleolar 3* (*GNL3*)	unsure	promote	Conservative stem cell gene, promotes regeneration	[72]
*Notum*	Wnt signaling	promote	Promotes the regeneration of aging tissues	[73]
*Oct-3/4, Sox2, Klf4 and c-Myc* (*OSKM*)	unsure	promote	Short-term induction of OSKM in muscle fibers can promote tissue regeneration by changing the niche of stem cells	[74]
*Early growth response* (*EGR*)	Jun N-terminal kinase (JNK) signaling	promote	Whole-body regeneration “switch”	[75]
*Equinox*	BMP signaling	promote	Promotes the formation blastema formation in regeneration	[76]
*HOX*	Wnt signaling	promote	Specifies the digestive system tissues	[77]
*FoxA*	Wnt signaling	promote	Specifies the tissues of the digestive system	[78]
*Bone morphogenetic protein 2* (*Bmp2*)	BMP signaling	promote	Enhances bone regeneration	[79]
*Bmp4*	BMP signaling	promote	Initiates regeneration	[76]

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
