# Peer review of "Comparisons between Plant and Animal Stem Cells Regarding Regeneration Potential and Application"

_ijms, 2023, doi:10.3390/ijms24054392_

Round 1

Reviewer 1 Report

This review article purports to summarize the state of knowledge of regeneration in plants and animals and provide comparisons and insights.  Unfortunately I believe that it fails to accomplish these goals.  My main criticisms are:

1)       It is not a review of regeneration but of stem cell maintenance.  The authors state that regeneration requires stem cells and then talk about stem cell niches and their role in development for most of the paper.  Regeneration, in contrast, is the regrowth of tissues that have been lost or damaged (or experimentally removed).  Almost no examples of regeneration experiments and their results, or our understanding of regeneration mechanisms, are mentioned in this review.

2)      It is vague or inaccurate in many many places.  Only some of this can be attributed to difficulties with English usage.  I will use the first paragraph of the introduction as an example, but vague generalizations and inaccuracies occur throughout the document. 

“Both lower and higher plants have strong regeneration capacities.”  What does this mean?  “Strong capacities” is not a term with scientific meaning as far as I know.  The authors use this repeatedly to compare organisms.  They also use the term “weak proliferative and replicative ability” to compare tissues – I again do not know what this means.  Another example on the next page:  “The regenerative ability of plants is generally stronger than that of animals, but also vary greatly between species. For example, the regenerative capacities of Taxus chinensis, Metasequoia glyptostroboides, and Ginkgo biloba is relatively weak, whereas those 110 of lower plants, such as Ficus virens, Laminaria japonica, and Undaria pinnatifida, are relatively strong.”  No reference is given for this statement.

“Tissue regeneration refers to the process by which tissues of organisms are continuously renewed, or are repaired via proliferation of surrounding healthy cells after damage occurs (2).”  This is not true – in morphollaxis, existing adult tissues redifferentiate to create new organs and there is very little proliferation involved. 

“Humans, however, can only regenerate intestinal cells, skin, hair, and bones either continuously or periodically (12, 13).”  This is not true.  Humans can regenerate fingertips and entire lobes of the liver.  In contrast, hair and fingernails are not living tissues and therefore do not qualify as examples of regeneration.

3)      There is no useful insights into parallels between plants and animal regeneration mechanisms.  Given the ancient evolutionary separation of plants and animals, it is not surprising that different genes are involved in regulating stem cell maintenance (and indeed regeneration).  However, one might look for conceptual similarities, where for example a shared problem has been solved in the same way, even through the mechanism is different.  One example is the patterning of a blastema in limb regeneration, which might be compared to the patterning of callus tissue produced after wounding in plants. 

Instead, in the section about plant/animal similarities, the authors say that both can be thought of at the cell/tissue and organismal scale, but this is not really an insight.  They then make a point about polarity, saying that heads/shoots can only be regenerated from one end of the organism and tails/shoots from the other.  However, in plants leaf tissues can be induced to regenerate roots, which seems to argue against an inherent polarity of all structures.  They also make the following comparison: “The stem cell niches of plants are stem tip stem cell niches and root tip stem cell while animals have different stem cell niches, such as the common hematopoietic stem cell niches (40).”  Plants also have shoot axial meristems and the root pericycle is believed to function as a meristem for the formation of lateral roots.  The stem cell niches of all animals is tremendously varied and can’t be summed up usefully in this way.  What is the point being made here?  Figure 2 seems to sum this all up by suggesting that the only similarities are that stem cells are involved in regeneration in both plants and animals.

Figure 1 is confusing to me.  I’m not sure what leads the authors to the conclusion about different regenerative capacities of “vegetative vs conducting vs protective” tissues in plants, nor of “lower plants” like ferns vs angiosperms and gymnosperms.  I don’t see a comparison made between left and right sides of the figure, or the significance of any of the triangles.  It’s pretty but I can’t figure out what it is meant to mean. 

4)       The mechanisms section is overly simplified and condensed to the point where it does not serve as a helpful starting point or review for people either skilled or new to the field.  It is at the same time highly detailed and extremely general.  For example, the following sentence comes from the single paragraph about notch signaling: “ Notch protein is cleaved three times, and its intracellular domain (NICD) is 274 released into the cytoplasm and enters the nucleus to bind to the transcription factor CBF- 275 1, suppressor of hairless, Lag (CSL) to form a transcriptional activation complex, thereby, 276 activating Hairy Enhancer of Split (HES), Hairy and Enhancer of split related with YRPW 277 motif (HEY), homocysteine-induced ER protein, and other basic helix-loop-helix (bHLH) 278 transcription factor family of target genes (116, 117).”  This level of detail is only appropriate if these proteins are explained.

Also, since no parallels have been drawn between plants and animals, it is really two distinct overly condensed summaries of stem cell maintenance with no integration (as stated above and shown in figure 2).

5)       The applications section is a discussion of very disparate ideas including the use of tissue culture to harvest metabolites, immortalized cell lines for artificial meat, organ regeneration, stem cell transplantation, antiviral mechanisms.  The ideas are only slightly explained and have no obvious connection beyond the idea that they are based on cells with stem cell–like characteristics.  There are also curious sentences which are perhaps typos i.e. . “The biological basis of shoot tip virus-free technology is that there is little or no SAM virus in plants. “

In summary, I became greatly frustrated in reading this article.  Terms were very poorly defined so that I could not figure out what points the authors were making.  The seemed to be little insight into the concept of regeneration – the points that were made were either quite shallow or were not supported by evidence.

I would recommend that these authors carefully define their goal in summarizing this complicated topic.  Perhaps setting out to define mechanisms of stem cell maintenance would be a better starting point for this article.  Then, discussing similarities and differences between the many different types of stem cells that exist in plants and animals, and their different potencies, would be instructive.  What determines pluripotency vs totipotency?  Can we compare stem cells in development vs in regeneration?  These are complex and fascinating concepts and I don’t doubt that these authors have important things to say about them.  From the starting point of stem cells, the technology review that the author offer would become quite interesting and make much more sense.  This article, unfortunately, as written fails to convey coherent ideas and contributes very little to the current literature.

Author Response

Response to Reviewer 1 Comments

Dear Reviewer:

Thank you for your comments on our manuscript entitled " Similarities and differences of plant and animal regeneration and progresses in practical applications" (ijms-2148652). Those comments were very helpful for improving our manuscript. We believed we have answered reviewer’s all comments. The main corrections in the paper and the responds to the reviewer’s comments are as follows:

Comments to the Author

This review article purports to summarize the state of knowledge of regeneration in plants and animals and provide comparisons and insights. Unfortunately I believe that it fails to accomplish these goals.  My main criticisms are:

Comment 1: It is not a review of regeneration but of stem cell maintenance.  The authors state that regeneration requires stem cells and then talk about stem cell niches and their role in development for most of the paper.  Regeneration, in contrast, is the regrowth of tissues that have been lost or damaged (or experimentally removed).  Almost no examples of regeneration experiments and their results, or our understanding of regeneration mechanisms, are mentioned in this review.

Response: Thank you for your suggestion. According your advice, we have replaced the title “Similarities and differences of animal and plant regeneration and progresses in practical applications” with “Comparisons between plant and animal stem cell regarding regeneration potential and application”.

At the same time, we have added examples and results of regeneration experiments, as well as understanding of regeneration mechanism in the manuscript.

We have added “The regeneration process includes tissue repair, de novo organ regeneration, the formation of wound-induced callus and somatic embryogenesis. Root tip repair involves the wounding response, the redistribution of auxin and cytokinin, and reconstruction of the quiescent center (QC) and stem cell niche re-establishment [94].” in the third paragraph of the third part in lines 184-187.

We have added “De novo root regeneration is the process by which adventitious roots form from wounded or detached plant organs. Auxin is the key hormone that controls root organogenesis and it activates many key genes involved in cell fate transition during root primordium establishment [102]. The detached leaves of Arabidopsis thaliana can regenerate adventitious roots on hormone-free medium [103]. From 10min to 2h after leaf detachment, a wave of JA is rapidly produced in detached leaves in response to wounding, but this wave disappears by 4h after wounding [104]. JA activates the expression of transcription factor gene ERF109 through its signaling pathway, which in turn up-regulates the expression of ANTHRANILATE SYNTHASE α1 (ASA1). ASA1 is involved in the biosynthesis of tryptophan, a precursor of auxin production. After 2h, the concentration of JA decreased, resulting in the accumulation of JAZ protein, which could directly interact with ERF109 and inhibit ERF109, thus turn off the wound signal. In general, the post-injury JA peak promotes auxin production to promote root regeneration from the cuttings, and root organogenesis also requires a strict turning-off of JA signal [105].” in the fourth paragraph of the third part in lines 218-231.

We have added “Plants can undergo multiple regenerative processes after wounding to repair wounded tissues, form new organs, and produce somatic embryos [109]. Plant somatic embryogenesis refers to the process in which somatic cells produce embryoids through in vitro culture [110]. This process can occur directly from the epidermis, sub-epidermis, cells in suspension, protoplasts of explants, or from the outside or inside of a callus formed from dedifferentiated explants. The transformation from somatic cells to embryogenic cells is the premise of somatic embryogenesis. In this process, the isolated plant cells undergo dedifferentiation to form a callus, the callus and cells undergo redifferentiation into different types of cells, tissues and organs, and finally generate complete plants [111]. This process involves cell reprogramming, cell differentiation and organ development, and is regulated by several transcription factors and hormones [112]. For example, the WUS gene regulates the transformation of auxin-dependent vegetative tissues to embryonic tissues during somatic embryogenesis [113,114]. Overexpression of WUS can induce somatic embryogenesis and shoot and root organogenesis. Ectopic expression of the WUS gene can dedifferentiate recalcitrant materials that do not undergo somatic embryogenesis easily, to produce adventitious buds and somatic embryos [115]. Additionally, LEAFY COTYLEDON 1 (LEC1), highly expressed in embryogenic cells, somatic embryos, and immature seeds, can promote somatic cell development into embryogenic cells. Furthermore, LEC1 can maintain the fate of embryogenic cells at the early stage of somatic embryogenesis. At present, LEC1 is used as a marker gene for somatic embryogenesis in several species [116]. Unlike LEC1, LEC2 can directly induce the formation of somatic embryos, which may activate different regulatory pathways [117].” in the sixth paragraph of the third part in lines 247-268.

Comment 2: It is vague or inaccurate in many many places.  Only some of this can be attributed to difficulties with English usage.  I will use the first paragraph of the introduction as an example, but vague generalizations and inaccuracies occur throughout the document.

“Both lower and higher plants have strong regeneration capacities.”  What does this mean?  “Strong capacities” is not a term with scientific meaning as far as I know.  The authors use this repeatedly to compare organisms.  They also use the term “weak proliferative and replicative ability” to compare tissues – I again do not know what this means.  Another example on the next page: “The regenerative ability of plants is generally stronger than that of animals, but also vary greatly between species. For example, the regenerative capacities of Taxus chinensis, Metasequoia glyptostroboides, and Ginkgo biloba is relatively weak, whereas those 110 of lower plants, such as Ficus virens, Laminaria japonica, and Undaria pinnatifida, are relatively strong.”  No reference is given for this statement.

“Tissue regeneration refers to the process by which tissues of organisms are continuously renewed, or are repaired via proliferation of surrounding healthy cells after damage occurs (2).”  This is not true – in morphollaxis, existing adult tissues redifferentiate to create new organs and there is very little proliferation involved.

“Humans, however, can only regenerate intestinal cells, skin, hair, and bones either continuously or periodically (12, 13).”  This is not true.  Humans can regenerate fingertips and entire lobes of the liver.  In contrast, hair and fingernails are not living tissues and therefore do not qualify as examples of regeneration.

Response: Thank you for your suggestion. We have modified these parts, we deleted the inappropriate parts in the new manuscript. We have deleted “Regeneration capacity can be considered as the replicative and proliferative abilities of cells within tissues. Higher animals and plants usually have complex structures, and tissues/organs are composed of terminal or highly differentiated cells with distinct specific functions. Cells that are highly differentiated have weak proliferative and replicative ability” in the first paragraph of the second part.

We have also added the reference supporting this statement. Reference (39): Perez-Garcia, P.; Moreno-Risueno, M.A. Stem cells and plant regeneration. Dev Biol 2018, 442, 3-12, doi: 10.1016/j.ydbio.2018.06.021.

We have replaced “Tissue regeneration refers to the process by which tissues of organisms are continuously renewed, or are repaired via proliferation of surrounding healthy cells after damage occurs”. with “Tissue regeneration refers to the continuous renewal of biological tissues, or the re-differentiation of existing adult tissues to produce new organs, or the repair process after tissue damage. It is one of biological life phenomena”. In lines 35-37.

And we apologize for the confusion in the text. We have modified this part, and deleted the “hair” which is not living tissue in line 45. See the highlighted yellow part in the manuscript for specific modifications. Correspondingly, we modified the picture and uploaded a new one.

Comment 3: There is no useful insights into parallels between plants and animal regeneration mechanisms.  Given the ancient evolutionary separation of plants and animals, it is not surprising that different genes are involved in regulating stem cell maintenance (and indeed regeneration).  However, one might look for conceptual similarities, where for example a shared problem has been solved in the same way, even through the mechanism is different.  One example is the patterning of a blastema in limb regeneration, which might be compared to the patterning of callus tissue produced after wounding in plants.

Instead, in the section about plant/animal similarities, the authors say that both can be thought of at the cell/tissue and organismal scale, but this is not really an insight.  They then make a point about polarity, saying that heads/shoots can only be regenerated from one end of the organism and tails/shoots from the other.  However, in plants leaf tissues can be induced to regenerate roots, which seems to argue against an inherent polarity of all structures.  They also make the following comparison: “The stem cell niches of plants are stem tip stem cell niches and root tip stem cell while animals have different stem cell niches, such as the common hematopoietic stem cell niches (40).”  Plants also have shoot axial meristems and the root pericycle is believed to function as a meristem for the formation of lateral roots.  The stem cell niches of all animals is tremendously varied and can’t be summed up usefully in this way.  What is the point being made here?  Figure 2 seems to sum this all up by suggesting that the only similarities are that stem cells are involved in regeneration in both plants and animals.

Figure 1 is confusing to me.  I’m not sure what leads the authors to the conclusion about different regenerative capacities of “vegetative vs conducting vs protective” tissues in plants, nor of “lower plants” like ferns vs angiosperms and gymnosperms.  I don’t see a comparison made between left and right sides of the figure, or the significance of any of the triangles.  It’s pretty but I can’t figure out what it is meant to mean.

Response: Thank you for your suggestion, we have deleted the no useful insights into parallels between plant and animal regeneration mechanisms, and we added the comparison between plant callus and animal blastema. Following “in both plants and animals, injury is a stimulus for the formation of specialized wound tissue that initiates regeneration. A regenerative response from these organisms can be elicited by environmental insults, even predatory or pathogenic attacks. Amputation in animals is usually but not always followed by the formation of a specialized structure known as a regeneration blastema. This structure consists of an outer epithelial layer that covers mesodermally derived cells and essentially defines a canonical epithelial/mesenchymal interaction, a conserved tissue relationship that is central to the development of complex structures in animals. In plants, one frequent but not universal feature of regeneration is the formation of a callus, a mass of growing cells that has lost the differentiated characteristics of the tissue from which it arose. A callus is typically a disorganized growth that arises on wound stumps and in response to certain pathogens. One common mode of regeneration is the appearance of new meristems within callus tissue. Thus, the plant callus shares with animal regeneration blastemas the property of being a specialized and undifferentiated structure capable of giving rise to new tissues”. in the second paragraph of the second part in lines 82-96.

Most of the regeneration of animals and plants have polarity characteristics, but there are also some special cases. Therefore, we have modified this part and deleted “Secondly, the regeneration of both animals and plants has polarity. If the head of planarian is removed, it can only regenerate the head upward, but not downward. However, if the tail of planarian is removed, the tail can only regenerate downward, but not upward. Since notum and wnt1 are limited to anterior and posterior wounds, respectively, and become highly expressed in their respective regions if that body part is removed. Similarly, Arabidopsis thaliana can only regenerate buds (but not roots) after its buds are cut off. The direction of regeneration is upward, and the same is true for roots. Animal and plant regeneration requires stem cell homeostasis, and the directional differentiation of stem cells requires a relatively fixed, regulatory active microenvironment, called niche. The niche of stem cells is composed of tissue cells and extracellular matrix, which controls the self-renewal of stem cells. The size of the niche is often limited by the source of signal molecules and the effective range they can reach. The stem cell niches of plants are stem tip stem cell niches and root tip stem cell niches, while animals have different stem cell niches, such as the common hematopoietic stem cell niches” in the second paragraph of the second part.

Due to our negligence in literature reading, we made mistakes in drawing C and D in the original drawing, so we decided to delete them, we have the deleted the Figure1 C and Figure1D.

The regeneration capacity of different plants is similar, so trapezoid is used to represent. However, the regenerative capacity of different animals varies greatly, so it is represented by a triangle. From top to bottom, the regeneration capacity increases in turn.

Comment 4: The mechanisms section is overly simplified and condensed to the point where it does not serve as a helpful starting point or review for people either skilled or new to the field.  It is at the same time highly detailed and extremely general.  For example, the following sentence comes from the single paragraph about notch signaling: “ Notch protein is cleaved three times, and its intracellular domain (NICD) is 274 released into the cytoplasm and enters the nucleus to bind to the transcription factor CBF- 275 1, suppressor of hairless, Lag (CSL) to form a transcriptional activation complex, thereby, 276 activating Hairy Enhancer of Split (HES), Hairy and Enhancer of split related with YRPW 277 motif (HEY), homocysteine-induced ER protein, and other basic helix-loop-helix (bHLH) 278 transcription factor family of target genes (116, 117).”  This level of detail is only appropriate if these proteins are explained.

Also, since no parallels have been drawn between plants and animals, it is really two distinct overly condensed summaries of stem cell maintenance with no integration (as stated above and shown in figure 2).

Response: Thank you for your suggestion, we have added some details about these signalings. We have added “β-catenin is a multifunctional protein, which helps cells respond to extracellular signals and influences by interacting with cytoskeleton (125).” in lines 286-287.

We have added “Dvl receives upstream signals in the cytoplasm and is the core molecule regulating Wnt signal pathway.” in lines 288-289. “TCF/LEF transcription factor meeting β-catenin protein, which initiates the expression of key genes in multiple Wnt signaling pathways.” in lines 295-296.

 We have added “There are four kinds of Notch receptors (Notch1-4) in mammals, which are composed of three parts: extracellular domain (NEC), transmembrane domain (TM) and intra-cellular domain (NICD).” in lines 321-323.

We have added “CSL protein is a key transcriptional regulator in Notch signaling pathway, which is also known as the classical Notch signaling pathway and CSL-dependent pathway”. in lines 325-327.

We have added “Hh signal transmission is controlled by two receptors on the target cell membrane, Patched (Ptc) and Smoothened (Smo). The receptor Smo is encoded by the proto-oncogene Smothened and is homologous to the G-protein-coupled receptor. It is composed of a single peptide chain with seven transmembrane regions. The N-terminal is located outside the cell, and the C-terminal is located inside the cell. The amino acid sequence of the transmembrane region is highly conserved (137). The serine and threonine residues at the C-terminal are phosphorylated sites. When protein kinase catalyzes, it binds phosphate groups. The members of this protein family have the function of transcription promoter only when they maintain their full length, and start the transcription of downstream target genes; When the carboxyl end is hydrolyzed by the proteasome, a transcription inhibitor is formed to inhibit the transcription of downstream target genes. Smo is a necessary receptor for Hh signal transmission”. in lines 341-353.

In addition, we have previously summarized that there are few similarities between animal and plant regeneration, but according to your suggestion, we have discussed the research related to regeneration in combination with stem cells. At the same time, we also have added the similarity comparison between animal bud and callus you mentioned, as well as the related content about somatic embryos.

Comment 5: The applications section is a discussion of very disparate ideas including the use of tissue culture to harvest metabolites, immortalized cell lines for artificial meat, organ regeneration, stem cell transplantation, antiviral mechanisms.  The ideas are only slightly explained and have no obvious connection beyond the idea that they are based on cells with stem cell–like characteristics.  There are also curious sentences which are perhaps typos i.e. . “The biological basis of shoot tip virus-free technology is that there is little or no SAM virus in plants.

Response: Thank you for your suggestion, the previous application part is based on the description of stem cells, which lacks the application of regeneration. We have added the “Regenerating adventitious roots from cuttings is a common plant clonal reproduction biotechnology in forestry and horticulture industries. Plant somatic embryogenesis also has broad application prospects in artificial seeds, haploid breeding, asexual reproduction and germplasm conservation [160].” in the second paragraph of the fourth part in lines 427-430.

We have also replaced the “The biological basis of shoot tip virus-free technology is that there is little or no SAM virus in plants.” with “Stem cells and their daughter cells from SAM of Arabidopsis thaliana can inhibit the infection of cucumber mosaic virus (CMV).” We also replaced the “artificial meat” with “cultured meat” and replaced the “soles” with “joints”.

At the same time, we have a more detailed description of the discussion of the application part. We have added the “In 2013, Dutch biologist Mark Post produced the first piece of artificial meat in history by using animal cell tissue culture method, which attracted widespread attention [165]. Animal cell culture artificial meat is mainly composed of skeletal muscle containing different cells. These skeletal muscle fibers are formed by the proliferation, differentiation and fusion of embryonic stem cells or muscle satellite cells. They first isolated the growth-differentiable primitive stem cells. By adding a culture medium rich in amino acids, lipids and vitamins, they accelerated cell proliferation and differentiation, and obtained a large number of bovine muscle tissue cells [166].” in the second paragraph of the fourth part in lines 447-455.

Comment 6: In summary, I became greatly frustrated in reading this article.  Terms were very poorly defined so that I could not figure out what points the authors were making.  The seemed to be little insight into the concept of regeneration – the points that were made were either quite shallow or were not supported by evidence. I would recommend that these authors carefully define their goal in summarizing this complicated topic.  Perhaps setting out to define mechanisms of stem cell maintenance would be a better starting point for this article.  Then, discussing similarities and differences between the many different types of stem cells that exist in plants and animals, and their different potencies, would be instructive.  What determines pluripotency vs totipotency?  Can we compare stem cells in development vs in regeneration?  These are complex and fascinating concepts and I don’t doubt that these authors have important things to say about them.  From the starting point of stem cells, the technology review that the author offer would become quite interesting and make much more sense.  This article, unfortunately, as written fails to convey coherent ideas and contributes very little to the current literature.

Response: Thank you for your comment. As for your summary, we very much agree that this manuscript mainly focuses on stem cells, and there are few descriptions about regeneration and no profound opinions. Based on your above summary, we have made major changes to the article. First, we have revised the title to further reflect the starting point of stem cells. Secondly, we supplemented the mechanisms and applications related to regeneration, such as root tip repair, root regeneration from scratch and somatic embryogenesis. Again, we have supplemented the relatively superficial description in the article, and cited references as evidence support. At last, we revised and deleted the inaccuracies in the article, and at the same time, we revised the language of the manuscript by professionals who are native to English (Editage), which made the article comprehensively improved.

Reviewer 2 Report

The authors have taken on an interesting and difficult topic to address in this review, specifically what is common and what is different between plant and animal regeneration. Because multicellularity was “created” independently in the plant and animal lineages, similarities may be expected to be rare. However, some “tools” were available in the single cell lineages prior to the split between plants and animals and could be recruited in similar processes to drive regeneration (or as I comment below, much of normal growth). Two such proteins mentioned are TOR and RBR, although how they share mechanisms of de- and redifferentiation is not clear. If I understand correctly, these proteins are involved in control of the cell cycle and thus would be expected to be involved in processes that require cell proliferation. I believe that cell death is also involved in true plant regeneration, but this did not come out in the text to compare to the comment in animals (line 235).

Plants overall have higher regeneration capacities than animals, where the ability to reform larger body parts is limited to “lower” animals, only occurring in some particular tissues in mammals for instance. While I agree this is true, I feel that the authors spend much of the time discussing what I would consider normal development in plants rather than dedifferentiation and establishment of different cell types by redifferentiation (regeneration). They discuss the molecular biology of shoot and root apical meristems, but I would argue that this is just part of normal development that occurs largely after embryogenesis (e.g., post-embryonic growth and development from meristems that were set up during embryogenesis). While some of the genes they discuss do have roles in (what I would consider) true regeneration (i.e., organogenesis and somatic embryogenesis, with WUS being one such factor) they do not really discuss this other than some comments on callus development and wound healing. So in summary, are they really looking at regeneration in plants, or rather just the fact that plants have much development from the meristems after germination. Meanwhile organ systems, tissue types, etc. are set up during embryogenesis in higher animal development.

Figure 1. Can the text be enlarged for easier reading. Also, if I understand what they are saying about 1c, the top of the inverted triangle represents species with limited regeneration capacity, whereas the point is those organisms with more extensive regeneration. If so, why are birds positioned as higher capacity than fish and amphibians? Also why is the image for reptile, what seems to be a crab? I also wondered what is known about animals that undergo metamorphosis. Also, in 1A, they are referring to differentiated tissues (fruits) as “vegetative tissue”, when in fact this would be reproductive phase of development. Vegetative tissue normally refers to the plant PRIOR to the transition to reproductive development (inflorescence meristem, floral meristem, flower development and seed/fruit).

Figure 2 – there are some typos: cryopreserved and meat for two. Also, above the top dotted line, I think they are showing me a fertilized egg, generating a heart stage embryo, but then the next arrow goes to callus with organ formation (organogenesis, true plant regeneration) rather than to say a dry seed and then a plant. While it is not incorrect that embryo explants can be used to generate callus that can then form organs, I think to parallel what they are showing in animals, they should at least have an arrow from the heart stage embryo to the plant (they could still have the second route by callus but need to explain).

Minor – some examples:

Line 3 – progress in practical application

Line 378 – I would say “lab-grown meat” or “cultured meat” since “artificial meat” can mean vegetarian “meat” (soy burgers).

Line 389 – “soles” of horses or do they mean “joints”

Lines 397 to 400 -  they discuss beauty products but I believe that some important medicines are produced in cell cultures (taxol…).

Lines 465-469 – as written, it is confusing. A context where plant regeneration would be used to propagate rare, precious, or plants that take many years to form seeds (trees) would be using somatic embryogenesis and encapsulating the somatic embryos into synthetic seeds. As it reads, it sounds like they are transforming plants with genes that would enable more seed production (in fairness, this could also require regeneration by somatic embryogenesis or organogenesis, but this is not  clarified).

In summary, they have picked a fascinating and hard to address topic, but I don’t think they are really discussing plant regeneration for the most part (I don’t think I even saw a discussion of somatic embryogenesis). I cannot comment on the animal summary as well.

Author Response

Response to Reviewer 2 Comments

Dear Reviewer:

Thank you for your comments on our manuscript entitled " Similarities and differences of plant and animal regeneration and progresses in practical applications" (ijms-2148652). Those comments were very helpful for improving our manuscript. We believed we have answered reviewer’s all comments. The main corrections in the paper and the responds to the reviewer’s comments are as follows:

Comments to the Author

The authors have taken on an interesting and difficult topic to address in this review, specifically what is common and what is different between plant and animal regeneration. Because multicellularity was “created” independently in the plant and animal lineages, similarities may be expected to be rare. However, some “tools” were available in the single cell lineages prior to the split between plants and animals and could be recruited in similar processes to drive regeneration (or as I comment below, much of normal growth). Two such proteins mentioned are TOR and RBR, although how they share mechanisms of de- and redifferentiation is not clear. If I understand correctly, these proteins are involved in control of the cell cycle and thus would be expected to be involved in processes that require cell proliferation. I believe that cell death is also involved in true plant regeneration, but this did not come out in the text to compare to the comment in animals (line 235).

Response: Thank you for your comment. According your suggestion, we have added “In plants, programmed cell death (PCD) plays crucial roles in vegetative and reproductive development (dPCD), and in the response to environmental stresses (ePCD) [120,121]. Sexual reproduction in plants is important for population survival and for increasing genetic diversity. During gametophyte formation, fertilization, and seed development, there are numerous instances of developmentally regulated cell elimination, several of which are forms of dPCD essential for successful plant reproduction [122].” in the paragraph 7 of section 3 (L272-L278).

We also have added “Cell growth and cell cycle progression are generally tightly connected, allowing cells to proliferate continuously while maintaining their size. TOR is an evolutionarily con-served kinase that regulate both cell growth and cell cycle progression coordinately [27].” in the paragraph 2 of section 1 (L59-L62).

And we have added “Like in animals, RBR in plants inhibits cell cycle progression by interacting with E2F transcription factor homologues. In addition, decreased RBR levels lead to increased numbers of stem cells, while increased RBR levels lead to stem cell differentiation, indicating that RBR plays an important role in stem cell maintenance.” in the paragraph 3 of section 3 (L210-L214).

Comment 1: Plants overall have higher regeneration capacities than animals, where the ability to reform larger body parts is limited to “lower” animals, only occurring in some particular tissues in mammals for instance. While I agree this is true, I feel that the authors spend much of the time discussing what I would consider normal development in plants rather than dedifferentiation and establishment of different cell types by redifferentiation (regeneration). They discuss the molecular biology of shoot and root apical meristems, but I would argue that this is just part of normal development that occurs largely after embryogenesis (e.g., post-embryonic growth and development from meristems that were set up during embryogenesis). While some of the genes they discuss do have roles in (what I would consider) true regeneration (i.e., organogenesis and somatic embryogenesis, with WUS being one such factor) they do not really discuss this other than some comments on callus development and wound healing. So in summary, are they really looking at regeneration in plants, or rather just the fact that plants have much development from the meristems after germination. Meanwhile organ systems, tissue types, etc. are set up during embryogenesis in higher animal development.

Response: Thank you for your suggestion. We have added a paragraph about somatic embryogenesis “Plants can undergo multiple regenerative processes after wounding to repair wounded tissues, form new organs, and produce somatic embryos [110]. Plant somatic embryogenesis refers to the process in which somatic cells produce embryoids through in vitro culture [111]. This process can occur directly from the epidermis, sub-epidermis, cells in suspension, protoplasts of explants, or from the outside or inside of a callus formed from dedifferentiated explants. The transformation from somatic cells to embryogenic cells is the premise of somatic embryogenesis. In this process, the isolated plant cells undergo dedifferentiation to form a callus, the callus and cells undergo redifferentiation into different types of cells, tissues and organs, and finally generate complete plants [112]. This process involves cell reprogramming, cell differentiation and organ development, and is regulated by several transcription factors and hormones [113]. For example, the WUS gene regulates the transformation of auxin-dependent vegetative tissues to embryonic tissues during somatic embryogenesis [114,115]. Overexpression of WUS can induce somatic embryogenesis and shoot and root organogenesis. Ectopic expression of the WUS gene can dedifferentiate recalcitrant materials that do not undergo somatic embryogenesis easily, to produce adventitious buds and somatic embryos [116]. Additionally, LEAFY COTYLEDON 1 (LEC1), highly expressed in embryogenic cells, somatic embryos, and immature seeds, can promote somatic cell development into embryogenic cells. Furthermore, LEC1 can maintain the fate of embryogenic cells at the early stage of somatic embryogenesis. At present, LEC1 is used as a marker gene for somatic embryogenesis in several species [117]. Unlike LEC1, LEC2 can directly induce the formation of somatic embryos, which may activate different regulatory pathways [118].” in the paragraph 6 of section 3 (L247-L268).

And the discuss about molecular biology of shoot and root apical meristems, because stem cells in plants are normally confined into stem cell niches (SCNs) within meristems. Meristems are proliferative zones in which tissues are generated from stem cells and grow. Because tissues are generated from stem cells, stem cells are also designated as tissue initials, although strictly tissue initials represent the origin of all distinctive tissue lineages. These tissue stem cells divide asymmetrically to regenerate themselves and form a stem cell daughter which proliferates, grow and eventually differentiates, so we writed the RAM and SAM.

About the role of genes in plant regeneration, we have added “At the genetic level, the highly specific and QC-expressed gene WOX5 delineates QC identity and maintenance [96]. WOX5 activity could most likely occur through direct effect on cell cycle regulators. Plants with disrupted expression levels of WOX5 show aberrant differentiation rates of the distal stem cells indicating the role of WOX5 in preventing stem cell differention [97].” in the paragraph 3 of section 3 (L196-L200). And we also have added “For example, the WUS gene regulates the transformation of auxin-dependent vegetative tissues to embryonic tissues during somatic embryogenesis [114,115]. Overexpression of WUS can induce somatic embryogenesis and shoot and root organogenesis. Ectopic expression of the WUS gene can dedifferentiate recalcitrant materials that do not undergo somatic embryogenesis easily, to produce adventitious buds and somatic embryos [116].” in the paragraph 6 of section 3 (L257-262).

Comment 2: Figure 1. Can the text be enlarged for easier reading. Also, if I understand what they are saying about 1c, the top of the inverted triangle represents species with limited regeneration capacity, whereas the point is those organisms with more extensive regeneration. If so, why are birds positioned as higher capacity than fish and amphibians? Also why is the image for reptile, what seems to be a crab? I also wondered what is known about animals that undergo metamorphosis. Also, in 1A, they are referring to differentiated tissues (fruits) as “vegetative tissue”, when in fact this would be reproductive phase of development. Vegetative tissue normally refers to the plant PRIOR to the transition to reproductive development (inflorescence meristem, floral meristem, flower development and seed/fruit).

Response: Thank you for your comment. We have modified these parts and enlarged the picture. We have adjusted the position of birds, fish and amphibians, and replaced crabs with tortoises as the representative of reptiles. In Figure 1A, it is indeed inappropriate to use fruit to represent "nutritional tissue". We also searched and combed the relevant contents in Figure 1A, and found that there was no significant difference in the regeneration ability between different tissues of plants, and it was inappropriate to use triangle representation, so we deleted Figure 1A. We have re-uploaded the modified Figure 1.

Comment 3: Figure 2 – there are some typos: cryopreserved and meat for two. Also, above the top dotted line, I think they are showing me a fertilized egg, generating a heart stage embryo, but then the next arrow goes to callus with organ formation (organogenesis, true plant regeneration) rather than to say a dry seed and then a plant. While it is not incorrect that embryo explants can be used to generate callus that can then form organs, I think to parallel what they are showing in animals, they should at least have an arrow from the heart stage embryo to the plant (they could still have the second route by callus but need to explain).

 Response: Thank you for your comment. We have replaced the “cryppreserved and artificial meat with cryopreserved and cultured meat”. In addition, we have added an arrow from the heart stage embryo to the plant. In the process of somatic embryogenesis, the development process from globular stage to torpedo stage is very similar to that of zygotic embryogenesis. Although there are many similarities in the morphology and cell process of zygotic embryogenesis and somatic embryogenesis, the mechanisms that determine the initiation of these two processes may be different. The specific characteristics of somatic embryogenesis may be that somatic embryogenesis originated from embryonic callus rather than fertilized eggs like zygotic embryogenesis. We have re-uploaded the modified Figure 2.

Minor – some examples:

Comment 4: Line 3 – progress in practical application

Response: Thank you for your comment. According to the suggestions of other reviewers, I further revised the title and replaced “applications” with “application” (L3).

Comment 5: Line 378 – I would say “lab-grown meat” or “cultured meat” since “artificial meat” can mean vegetarian “meat” (soy burgers).

Response: Thank you for your comment. We have replaced the “artificial meat” with “cultured meat” (L455 and L458).

Comment 6: Line 389 – “soles” of horses or do they mean “joints”

Response: Thank you for your comment. We have replaced the “soles” with “joints” (L462).

Comment 7: Lines 397 to 400 -  they discuss beauty products but I believe that some important medicines are produced in cell cultures (taxol…).

Response: Thank you for your suggestion. We have added the related content “In addition, Taxus chinensis and Catharanthus roseus suspension cell culture can also be used to produce taxol and vinblastine anticancer substances” in the paragraph 4 of section 4 (L472-L473).

Comment 8: Lines 465-469 – as written, it is confusing. A context where plant regeneration would be used to propagate rare, precious, or plants that take many years to form seeds (trees) would be using somatic embryogenesis and encapsulating the somatic embryos into synthetic seeds. As it reads, it sounds like they are transforming plants with genes that would enable more seed production (in fairness, this could also require regeneration by somatic embryogenesis or organogenesis, but this is not clarified).

Response: Thank you for your suggestion. We have clarified this point and added the sentence “Plant regeneration is mainly carried out through somatic embryogenesis or organo-genesis, but plant regeneration can be assisted by transferring plant-related genes (194)” in the paragraph 2 of section 5 (L537-L539).

Comment 9: In summary, they have picked a fascinating and hard to address topic, but I don’t think they are really discussing plant regeneration for the most part (I don’t think I even saw a discussion of somatic embryogenesis). I cannot comment on the animal summary as well.

Response: Thank you for your comment. We have added the contents related to plant regeneration (including the discussion of somatic embryogenesis), which is in the answer to comment 1.

Reviewer 3 Report

In this review article, the authors discussed an interesting, challenging, and topical topic in the field.

The article is correctly made, it has a logical presentation and a theoretical foundation, the synthesis of current knowledge about stem cells. The observations that I include in the analysis of this article are the following:

1). In the beginning section, the authors are asked to insert the objective of this article, the purpose of its writing.

2). The figures/diagrams inserted in this article must be homogenized in terms of their resolution.

3). The list of bibliographic references must be corrected from the point of view of the Instructions for Authors.

Author Response

Response to Reviewer 3 Comments

Dear Reviewer:

Thank you for your comments on our manuscript entitled " Similarities and differences of plant and animal regeneration and progresses in practical applications" (ijms-2148652). Those comments were very helpful for improving our manuscript. We believed we have answered reviewer’s all comments. The main corrections in the paper and the responds to the reviewer’s comments are as follows:

Comments to the Author

In this review article, the authors discussed an interesting, challenging, and topical topic in the field. The article is correctly made, it has a logical presentation and a theoretical foundation, the synthesis of current knowledge about stem cells. The observations that I include in the analysis of this article are the following:

Comment 1: In the beginning section, the authors are asked to insert the objective of this article, the purpose of its writing.

Response: Thank you for your suggestion. At the beginning of the article, we added the writing purpose of the manuscript and integrated the writing purpose with the third paragraph of the first part. We have replaced “Stem cells and their metabolites have great application value in agriculture, regenerative medicine and so on. Human health is one of the driving forces for the development of life sciences, and the rise of regenerative medicine is also serving human health (26). Regenerative medicine is promising for medical applications, stem cells are ideal seed cells for tissue engineering applications, stem cell culture can repair and regenerate tissues and organs, and overcome immune rejection. With over 80% gene homology with humans, the processes of planarian stem cells are highly similar to humans during the early stages post-injury (27). However, only a few human organs have repair capacity, and this capacity is generally very limited and decreases with age. Understanding how animals with strong regeneration capacities regulate regeneration may help identify methods to regenerate human organs and delay aging (28). This article reviews the molecular mechanisms and applications of plant and animal regeneration and provides new ideas for research and applications of plant and animal regeneration in agricultural production and medical treatment.” with “Stem cells and their metabolites have great application value in agriculture, regenerative medicine. Among them, the advancement in regenerative medicine benefits human health, and it has great prospects in the medical field [28]. Stem cells can be regarded as ideal seed cells for genetic engineering, able to the repair damaged tissues and organs, and to overcome immune rejection. In this review, we have discussed the regeneration mechanism of animals and plants, highlighting the similarities and differences between this biological process. Additionally, we summarize the main recent findings on animal and plant stem cells in the field of regeneration, and provide new ideas and directions for the protection of endangered species and the development of regenerative medicine.” in the paragraph 3 of section 1 in lines 63-71.

Comment 2: The figures/diagrams inserted in this article must be homogenized in terms of their resolution.

Response: Thank you for your comment. We have modified figures/diagrams and ensured that they maintain a uniform resolution. We uploaded the new high-resolution figures.

Comment 3: The list of bibliographic references must be corrected from the point of view of the Instructions for Authors.

Response: Thank you for your comment. We have checked all references and corrected them according to the instructions of the authors.

Round 2

Reviewer 1 Report

The authors have taken steps towards correcting the problems perceived by both this reviewer and reviewer 2.  However, I still find this article very difficult to read, both in terms of logic and in terms of English usage.  Initial reviews 1 and 2 both made the point that no real insights were provided into how plant and animal regeneration processes are similar.  Suggestions were made by the reviews as to how the authors might consider these questions in their revisions, with specific examples given.  The authors have responded by including paragraphs of details about each of the topics suggested by the reviewers.  In sum, the review contains highly condensed summaries of a numer of disparate topics without real attempts to integrate them into a single coherent piece.  Descriptions of transcription factor cascades in plant and animal stem cell maintenance, simplified to fit into small space, will not be useful for people unfamiliar with the field.  The mechanisms are not apparently similar (with the possible exception of the involvement of RBR and TOR).  Hence, while this review will not do any damage, I am not sure who the intended audience would be for such a review or how it would add to the literature.

One point that caught my interest but that I did not understand was the distinction made in figure 2 between products that could be derived from totipotent vs pluripotent vs unipotent stem cells.  The logic of this figure is not described in the text, and it seems like there might be an important point to make here.

Much of Figure 1 should be removed before publication as it is not explained well and does not reflect information contained in the review.  

Author Response

Response to Reviewer 1 Comments Dear Reviewer: Thank you for your comments on our manuscript entitled “Comparisons between plant and animal stem cell regarding regeneration potential and application” (ijms-2148652). Those comments were very helpful for improving our manuscript. We believed we have answered reviewer’s all comments. The main corrections in the paper and the responds to the reviewer’s comments are as follows: Comments to the Author: Comment 1:The authors have taken steps towards correcting the problems perceived by both this reviewer and reviewer 2. However, I still find this article very difficult to read, both in terms of logic and in terms of English usage. Initial reviews 1 and 2 both made the point that no real insights were provided into how plant and animal regeneration processes are similar. Suggestions were made by the reviews as to how the authors might consider these questions in their revisions, with specific examples given. The authors have responded by including paragraphs of details about each of the topics suggested by the reviewers. In sum, the review contains highly condensed summaries of a number of disparate topics without real attempts to integrate them into a single coherent piece. Descriptions of transcription factor cascades in plant and animal stem cell maintenance, simplified to fit into small space, will not be useful for people unfamiliar with the field. The mechanisms are not apparently similar (with the possible exception of the involvement of RBR and TOR). Hence, while this review will not do any damage, I am not sure who the intended audience would be for such a review or how it would add to the literature. Response: Thank you for your suggestion. In order to make the overall structure of the article clearer and more logical, we have invited professional English speakers to help us make language changes. Similarities concerning plant and animal regeneration, we have added “Moreover, the process of stem cell regeneration induced by somatic cells in plants is similar to that induced by animal pluripotent stem cells. In animals, the production of induced pluripotent stem cells(iPSC)depends on the expression of many key transcription factors. Similar to animal cells, the induction and maintenance of stem cells in plants also depend on the induction and expression of several key transcription factors such as class B-ARR, WUS or WOX5. Therefore, the stem cells induced in plants that express the pluripotent genes such as WUS or WOX5 can also be called plant-induced pluripotent stem cells (Plant iPSC) [40].” in the paragraph 2 of section 2 (L96-L104). We have deleted the uninterpreted gene “CYCD6;1” “RBR and SCR in RAM” in Figure 1A. Other genes are described in the article, in the specific description part, we have added “(Figure 1A)” in line 193, line 227 and line 259, and added “(Figure 1B)” in line 318, line 346, line 376 and line 401. At the same time, we have modified Figure 1 and re-uploaded it. Comment 2:One point that caught my interest but that I did not understand was the distinction made in figure 2 between products that could be derived from totipotent vs pluripotent vs unipotent stem cells. The logic of this figure is not described in the text, and it seems like there might be an important point to make here. Response: Thank you for your suggestion, about the applications of different stem cells of animals and plants, we have added “Plant totipotent stem cells have a good application prospect in crop breeding. The totipotent stem cells of animals in the placenta can be cryopreserved to treat some dis-eases after adulthood. Plant pluripotent stem cells and their metabolites can be used in the development of drugs, health foods and cosmetics. For animals, induced pluripotent stem cells can produce various organs needed, but at present, due to ethical constraints, artificial organs have not been allowed [166]. Artificial meat that can be made from animal multipotent stem cells can also be used for pet disease treatment. The unipotent stem cells of plants are also used for the extraction of some pigment sub-stances. In addition, the purple shirt stem cells in suspension culture can produce anti-cancer substances such as Taxamairin A and B, and the unipotent stem cells in milk have therapeutic potential in treating some animal diseases [169].” in the paragraph 2 of section 4 (L434-L444), and there are further explanations in other paragraphs of the fourth section. Comment 3:Much of Figure 1 should be removed before publication as it is not explained well and does not reflect information contained in the review. Response: Thank you for your suggestion. We have deleted the uninterpreted gene “CYCD6;1” “RBR and SCR in RAM” in Figure 1A. Other genes are described in the article, in the specific description part, we have added “(Figure 1A)” in line 193, line 227 and line 259, and added “(Figure 1B)” in line 318, line 346, line 376 and line 401. At the same time, we have modified Figure 1 and re-uploaded it. And we also separate the signal paths in Figure 2 with dotted lines to make readers read more clearly. We also replaced ARR1/12 in Figure 1A with ARR12, and added “AAR12 of cytokinin signal transduction pathway is the main enhancer of callus formation [117].” in lines 251-252.

Reviewer 2 Report

I think with the redirection from "regeneration" to the nature of stem cells, this article is a valuable contribution.

Author Response

Response to Reviewer 2 Comments Dear Reviewer: Thank you for your comments on our manuscript entitled “Comparisons between plant and animal stem cell regarding re-generation potential and application” (ijms-2148652). Those comments were very helpful for improving our manuscript. We believed we have answered reviewer’s all comments. The main corrections in the paper and the responds to the reviewer’s comments are as follows: Comments to the Author: I think with the redirection from "regeneration" to the nature of stem cells, this article is a valuable contribution. Response: Thank you for your guidance, which has greatly improved our article. We are very grateful.